# Dietary Fibre Modulates the Gut Microbiota

**DOI:** 10.3390/nu13051655

**Published:** 2021-05-13

**Authors:** Peter Cronin, Susan A. Joyce, Paul W. O’Toole, Eibhlís M. O’Connor

**Affiliations:** 1Department of Biological Sciences, University of Limerick, V94 T9PX Limerick, Ireland; Peter.Cronin@ul.ie; 2APC Microbiome Ireland, University College Cork, T12 K8AF Cork, Ireland; s.joyce@ucc.ie (S.A.J.); pwotoole@ucc.ie (P.W.O.); 3School of Biochemistry and Cell Biology, University College Cork, T12 K8AF Cork, Ireland; 4Department of Microbiology, University College Cork, T12 K8AF Cork, Ireland; 5Health Research Institute, University of Limerick, V94 T9PX Limerick, Ireland

**Keywords:** dietary fibre, metabolic health, microbiota, obesity, type II diabetes mellitus, metabolic syndrome, glucose metabolism, lipid metabolism

## Abstract

Dietary fibre has long been established as a nutritionally important, health-promoting food ingredient. Modern dietary practices have seen a significant reduction in fibre consumption compared with ancestral habits. This is related to the emergence of low-fibre “Western diets” associated with industrialised nations, and is linked to an increased prevalence of gut diseases such as inflammatory bowel disease, obesity, type II diabetes mellitus and metabolic syndrome. The characteristic metabolic parameters of these individuals include insulin resistance, high fasting and postprandial glucose, as well as high plasma cholesterol, low-density lipoprotein (LDL) and high-density lipoprotein (HDL). Gut microbial signatures are also altered significantly in these cohorts, suggesting a causative link between diet, microbes and disease. Dietary fibre consumption has been hypothesised to reverse these changes through microbial fermentation and the subsequent production of short-chain fatty acids (SCFA), which improves glucose and lipid parameters in individuals who harbour diseases associated with dysfunctional metabolism. This review article examines how different types of dietary fibre can differentially alter glucose and lipid metabolism through changes in gut microbiota composition and function.

## 1. Introduction

The human gastrointestinal tract (GIT) harbours one of the most complex ecosystems on the planet; a vast community of microbial residents exist here in an intricate relationship that has co-evolved over time. It is hypothesized that approximately 3.3 million microbial genes are encoded in the entire genetic repertoire of the gut microbiota [1]. In humans, one of the major influences on the microbial signatures of individuals is diet [2,3], while antibiotic use [4], exercise [5] and age [6] also have significant effects. Furthermore, the gut microbiota profiles of humans are altered in metabolic disease states such as obesity [7], type II diabetes mellitus (T2DM) [8] and cardiovascular disease (CVD) [9]. The emergence of the low-fibre Western diet is associated with an increased incidence of these metabolic disease states, particularly in industrialised nations [10,11]. Numerous methods are being developed to alter existing microbial populations in the GIT to deliver health benefits and disease prevention strategies, such as faecal microbial transplants (FMT) [12]. Dietary fibre intervention represents another method by which we can modify the microbiota to achieve improved health outcomes. Fibre has long been postulated as having an important role in the regulation of metabolism and the prevention of chronic gastroenterological diseases [13,14,15,16]. Recent advances in the field of microbiome science have unveiled the complexity of fibre’s relationship with human health and disease [17]. In fact, clear evidence emerging from both pre-clinical and clinical studies has indicated that different types of dietary fibre, or the metabolic products that are generated from their fermentation (e.g., short-chain fatty acids (SCFAs)), have extremely beneficial metabolic effects in the host [18]. SCFAs produced through fibre fermentation in the colonic region are known to positively regulate glucose and lipid metabolism [18]. By identifying how specific fibre types regulate host microbiota populations, and by further elucidating the mechanisms by which the fermentation of fibre by these microorganisms affects metabolism and gut homeostasis, a gateway of possibilities opens up for targeted therapeutic dietary fibre intervention to aid in alleviating certain disease states associated with altered metabolism. Thus, dietary fibre interventions may provide a solution in reducing the societal burden of gut diseases, such as obesity, metabolic syndrome and T2DM, on Western nations. This review aims to establish the effects that specific fibre types have on host microbial populations, and in turn what effect these changes have on host metabolism and disease pathology in metabolically challenged individuals. The research articles included in this literature review were identified and selected through PubMed. The search parameters included keywords such as “dietary fibre”, “microbiota”, “metabolic syndrome”, “obesity” and “type II diabetes”. Only the most recent articles were selected where possible.

## 2. What Is Dietary Fibre?

Dietary fibre is comprised of plant-based carbohydrates that cannot be metabolised by digestive enzymes encoded in the human genome, such as amylase. Instead, fibre can only be metabolized by certain species of gut microbiota through anaerobic fermentation, with the main product of this reaction being SCFAs. The specific definitions of fibre offered by different bodies has been reviewed elsewhere [19]. Two of the key differences that exist between the different organisations’ definitions of dietary fibre are related to the non-carbohydrate component of some types, as well as the number of monomers that constitute a fibre molecule [20,21]. In Europe, this number is as low as 3, while in other countries at least 10 monomeric units are required to define the polymer as a fibre [20]. Fibres with 10 or more monomeric units include non-starch polysaccharides (e.g., cellulose, hemicellulose, gums, pectin, mucilage, inulin, psyllium and β-glucan) and resistant starch (RS) (Figure 1) [17]. Fibres that are between three and nine monomeric units are called resistant oligosaccharides and include galacto-oligosaccharide (GOS) and fructo-oligosaccharide (FOS) (Figure 1) [22]. Although a universally accepted definition of dietary fibre does not exist, each type encompassed within the numerous definitions has a role in the regulation of gut microbiota composition and a subsequent effect on host metabolism. The specific actions exerted by fibre on the host depend on its physiochemical properties, such as solubility, viscosity and fermentation, which differ greatly between fibre types [23]. These factors are in essence what defines the role of different fibre types in regulating the microbiota and the subsequent effect this has on host health and disease. The solubility of fibre types refers to their dissolution capability in water, while their fermentability refers to the degree to which they can be metabolised by host microbes. Viscosity on the other hand is the ability of fibre to form a gel-like consistency in water. Specifically in relation to the microbiome, the term microbiota-accessible carbohydrate (MAC) has emerged, and this definition of fibre excludes the insoluble fibres mentioned, as they cannot be utilised by the gut microbiota. β-glucan, gums and pectin are all prime examples of highly soluble, fermentable and viscous fibres [24].

## 3. Dietary Fibre Regulates the Composition of the Gut Microbiota

Dietary fibre can have a major impact on the composition, diversity and richness of the microbiome, providing a plethora of substrates for fermentation reactions carried out by specific species of microbe that possess the necessary enzymatic machinery to degrade these complex carbohydrates. There are numerous microbiota species defined as fibre fermenters in the large intestine, while there are different types of dietary fibre, all of which broken down. Increasing dietary fibre consumption alters the nutritional niches in the intestine, allowing these bacteria to expand their populations [25]. Individuals consuming diets low in fibre tend to have reduced microbial diversity [21]. These individuals are likely to harbour microbes that thrive on amino acids and lipids, as the reduced dietary fibre is likely to be replaced in the diet by animal protein and fat. Numerous studies outline how the consumption of different diets, in different geographical locations and socio-economic backgrounds, can alter microbial populations in the human gastrointestinal tract [26,27,28,29,30,31,32,33,34]. One commonality between these studies is that people from less developed and rural societies consume a significantly larger portion of fibre in their diet as compared with individuals from industrialised nations [26,31]. Interestingly, metabolic and inflammatory diseases such as obesity and IBD have a very low prevalence in individuals from unindustrialised nations. 

De Filippo et al. (2010) [26] highlighted differences in the gut microbiome between Burkina Fasian and Italian children. Higher levels of *Actinobatceria*, *Bacteroidetes* and *Prevotella* were observed in Burkina Fasians while the Italian children had a higher abundance of *Firmicutes* and Proteobacteria [26]. An analogous study conducted by the same author measured the microbiota across four cohorts of differing urbanisation [34]. This study highlighted the reduced consumption of fibre among children living in urban Italy and urban Burkina Faso, which was further reflected by reduced *Prevotella* species in their microbiota [34]. Interestingly, the differences in microbiota reported here were almost identical to those reported by Ou et al. (2013) [28], who studied the microbiome of rural South Africans, comparing them with African Americans. 

In 2015, a study that investigated the microbiota of Papa New Guineans in comparison with Americans found that the faecal microbiota had an altered *Prevotella*:*Bacteroides* ratio [33]. In this study the microbiota of Papa New Guineans who consumed a diet rich in fibre had a high abundance of *Prevotella* but a low abundance of *Faecalibacterium*, *Ruminococcus*, *Bifidobacterium*, *Bacteroides*, *Blautia*, *Bilophila*, and *Alistipes* [33]. Yatsunenko et al. (2012) [27] reported similar findings when comparing the microbiota populations of individuals from Venezuela, Malawi and the USA. 

Clemente et al. (2015) [32] conducted a study investigating the microbiota of the Yanomami tribe in rural Venezuela. One of the key findings from this study was the higher abundance of *Prevotella* and lower abundance of *Bacteroides* among these people when compared with individuals from USA [32]. In addition, this population had a lower abundance of other members of the *Bacteroidetes* family, including *Bacteroidales*, *Mollicutes* and *Verrucomicrobia*. Lastly, these individuals were reported to have an increased abundance of the genera *Phascolarctobacterium*, which is known to produce SCFAs [35]. Another study similarly compared the microbiota of the Tanzanian Hadza tribe to Italians [30]. Both populations were found to contain an abundance of well-known fibre-degrading bacteria, including the *Firmicutes* family such as *Lachnospiraceae*, *Ruminococcaceae*, *Veillonellaceae*, *Clostridiales Incertae Sedis XIV* and *Clostridiaceae*. In keeping with the findings in other studies, *Prevotella* was enriched in the Hadza microbiome, while a lower abundance of *Bacteroides* was observed when compared with their Italian counterparts [30]. Another study comparing the effects of a fibre-rich, plant-based diet and a protein-rich animal-based diet on individuals from the USA showed that higher levels of *Prevotella* were linked to fibre degradation [29]. Furthermore, correlations were found between the plant-based diet and the taxa *Roseburia*, *Eubacterium rectale* and *Faecalibacterium prausnitzii* [29], as well as increased total concentrations of SCFAs. 

Thus, it is clear that the fibre component of the diet can influence the microbiota. An apparent trend exists in the microbiota of people from industrialised nations, and it appears that these individuals have a higher abundance of *Bacteroides*, *Bifidobacterium*, Ruminococcus, *Faecalibatcerium*, *Alistipies*, *Bilophila* and *Blautia* [26,27,30,33,34]. Previous comparisons of the microbiota between industrialised nations reveal little difference, highlighting that cultural and ethnic differences presumably exert little effect on the host microbiota [36]. Interestingly, these nations all consume a variation of the Western diet, which is high in saturated fat and low in fibre. Another key finding from these studies is that the *Prevotella* species is higher in abundance in non-industrialised individuals who consume vast quantities of dietary fibre. Interestingly, many species found in the microbiome of Westerners have been linked to the vast consumption of lipids and amino acids, which predominate in the Western diet [29,37]. The microbiota of non-industrialised populations are similar to those of vegans and vegetarians in Western society, all of whom consume vast quantities of dietary fibre [29]. High fibre consumption in combination with the specific species of microbe able to ferment it will lead to a range of health benefits for the host, including extensive SCFA production.

## 4. Different Species of Microbe Have the Capacity to Degrade Specific Fibre Types

An extensive range of microbial enzymes have been identified to participate in the degradation of dietary fibre in the large intestine. A specific fibre type may require multiple steps of enzymatic catalysis in order to yield an SCFA product. Thus, numerous microbes may be required in order for the host to benefit fully from these reactions. Some of these bacteria are highly specialised and make a major contribution to fibre degradation, thus being referred to as primary degraders or keystone species [38,39]. Moreover, other microbes have a minor role, and are referred to as secondary fermenters or cross-feeders as they benefit from the catalytic labour carried out by the primary degraders. Some members of the colonic microbial community, such as Bacteroides thetaiotaomicron, exhibit profound flexibility as they encode numerous enzymes that can contribute to the degradation of multiple fibre subtypes [40]. Some of the main bacteria-encoded enzymes involved in fermentation are referred to as carbohydrate-active enzymes (CAZymes) [41], which includes glycoside hydrolases, polysaccharide lysases and carbohydrate esterase [42]. 

Compositional differences between various types of resistant starch have been found to have differential effects on the host microbiota [43,44,45]. In vivo studies have found that RS4 consumption in humans increased the abundance of the phyla *Actinobacteria* and *Bacteroidetes*, while *Firmicutes* were decreased [46]. Furthermore, RS4 was shown to increase the abundance of *Parabacteroides distasonis* and *Bifidobacterium asolescentis*. In the same study, the dietary supplementation of RS2 had no effect on phylum, while at the species level population increases of *Ruminococcus bromii* and *Eubacterium rectale* were detected [46]. In vitro experimentation found that *Ruminococcus bromii* is a keystone species required for the fermentation of RS2 and RS3 [39]. The same author identified that while *Ruminococcus bromii* could not produce SCFAs of *Bifidobacterium asolescentis*, *Eubacterium rectale* and *Bacteroides thetaiotaomicron* could survive without it. This indicates that cross-feeding occurs, whereby *Ruminococcus bromii* carries out an initial degradation of RS, the product of which is further fermented by the other colonic microbes to produce SCFA [39]. Another report identified that *Bifidobacterium breve* and *Bifidobacterium adolescentis* encode enzymes that can also degrade resistant starch [47]. More recently, RS2 fed to healthy individuals increased the abundance of *Ruminococcus UCG-005* while reducing the populations of *Coprococcus*, *Bacteroiedes*, *Lachnoclostridium*, *Eubacterium eligens*, *Blautia*, *Holdemanella* and *Paraprevotella* (Table 1) [48].

Arabinoxylan, a well-known variety of hemicellulose, is thought to be degraded in the human colon by *Roseburia*, *Bacteroides*, *Prevotella* and *Porphyromonas* [49]. A human intervention using this fibre to modulate metabolic parameters in overweight humans identified increases in the abundance of *Prevotella* and *Eubacterium rectale* [50]. Another hemicellulose, xylan, was found to be fermented by several *Bacteroides* species [51,52]. Furthermore, *Roseburia intestinalis* was also identified to possess xylan-degrading capabilities [53]. 

The effects of increased inulin consumption have been investigated in vivo and were shown to increase the abundance of *Bifidobacterium bifidum* [54,55,56]. Inulin in humans has also been found to increase the abundance of *Faecalibacterium prausnitzii* (Table 1) [57], while lower levels of Enterococcus species were detected [58]. The supplementation of dietary inulin in humans gave rise to elevated populations of *Roseburia intestinalis*, *Eubacterium rectale* and *Anserostipes caccae* [59,60,61]. Interestingly, cross-feeding was observed by host microbes in the fermentation of inulin, similar to that reported for RS, and it appears that *Eubacterium rectale* is involved in the initial breakdown of inulin while *Anserostipes caccae* can ferment only smaller fragments of this molecule [62]. A fibre intervention in obese/overweight individuals consuming wheat bran led to increases in the phyla *Firmicutes* and *Bacteroidetes* [63]. Meanwhile, Salonen et al. (2014) [64] reported increases in the abundance of the *Actinobacteria* phylum, while at the genus level higher levels of *Prevotella*, *Bacteroides*, *Eggerthella*, *Lachnospiraceae*, *Corynebacterium* and *Collinsella* were reported in obese and metabolically challenged men also supplemented with wheat bran. Contrary to this, a whole-grain wheat intervention conducted on healthy individuals described increases in *Enterococcus*, *Bifidobacterium*, *Clostridium* and *Lactobacillus* [65], suggesting personalized effects relative to the initial resident microbiome. 

An oat derived β-glucan incorporated into the diet appears to lead to higher levels of *Bacteroidetes* and a decrease in the abundance of *Firmicutes* (Table 1) [66]. Experimental evidence in rats suggests that dietary intervention using this fibre type increases the abundance of *Bacteroides* and *Prevotella* species [67]. Elsewhere, it was reported that β-glucan causes increases in the abundance of *Bifidobacteria* [68] as well as *Bacteroides* species (Table 1) [69]. However, *Clostridium histolyticum* was also observed at higher levels [70], with this species of microbe being a well-known butyrate producer [71,72]. Interestingly, many other members of the *Clostridium* family encode β-glucanase enzymes, and thus have the capacity to ferment this fibre type [73,74,75].

A gum acacia fibre intervention in humans resulted in higher levels of *Bifidobacteria* and *Lactobacilli* [76]. Furthermore, it has also been suggested that one of the main fermenters of gum fibres is *Prevotella ruminocola* [77,78], while *Bacteroides* has been postulated to have a role [79]. Other species include *Propionibacterium*, *Veillonella*, *Selenomonas ruminatium* and *Anaerovibrio lypolytica* [77]. The short-chain fibre galacto-oligosaccharide (GOS) is related to higher levels of *Bifidobacterium* species, which are well equipped to ferment this complex carbohydrate [80]. Meanwhile, *Faecalibacterium prausnitzii* has been reported to increase in abundance with dietary GOS supplementation [80], while fructo-oligosaccharide (FOS), a short-chain fibre subtype similar to GOS, can be fermented by species of *Bifidobacterium* and *Collinsella aerofaciens* [81]. Thus, several dietary fibre fractions can positively impact gut microbial populations.

Increases in whole-grain barley intake has been reported to increase the abundance of *Prevotella copri* [82]. Another study investigating the effects of barley reported increases in the phyla *Firmicutes* and *Actinobacteria* [83]. At the species level, *Bifidobacterium*, *Roseburia*, *Eubacterium* and *Dialister* were the main responders to this intervention, which was accompanied by a decrease in the *Bacteroidetes* phylum. However, an investigation into whole-grain rye fibre demonstrated no effect on faecal microbiota in a healthy Danish cohort [84]. In a similar intervention, no alterations in the total abundance of gut microbiota of Finnish individuals were observed in those diagnosed with metabolic syndrome (Table 1) [85]. However, this author did report significantly lower levels of *Bacteroidetes* and higher levels of *Firmicutes* and *Actinobacteria* following supplementation with 75 g of rye bread per day. 

It is clear that dietary fibre has a profound effect on the composition of the gut microbiota, with the degradation of different types of fibre requiring numerous different species that encode complementary enzymes for fermentation. This syntrophic relationship sees primary degraders or keystone species initiate the process and produce partially broken-down products, in addition to metabolites, which can benefit the keystone species itself as well as other taxa [86]. Cross-feeding occurs as secondary fermenters complete the process by fully breaking down the dietary fibre to produce SCFAs [38,87]. This cross-feeding has been fully described for RS fermentation. Further studies are required to establish these relationships in the breakdown of other dietary fibre types. In order for the potential of dietary fibre to be fully utilised in the treatment of metabolic disorders, it is essential that the specific role these microbiota play in metabolising particular fibre subtypes is investigated.

## 5. SCFAs Are a Product of Microbial Fermentation of Dietary Fibre

The fermentation of dietary fibre by host microbiota has been associated with a plethora of health benefits, including lower cholesterol and improved glucose control [17]. In addition to having a considerable impact on the gut microbiota, fibre can also dictate the microbial-derived metabolite signatures in the intestine [88]. SCFAs are the main product of these microbial biochemical alterations, and include butyrate, acetate and propionate. Butyrate is produced through what is known as the classical pathway or the butyryl CoA:acetate CoA transferase pathway [89,90]. Acetate, on the other hand, is synthesised from pyruvate through the Wood–Ljungdahl pathway [91] or through acetyl CoA [92]. Lastly, propionate is produced through the succinate pathway, acrylate pathway or propanediol pathway [92,93,94]. Microbes can also alter these SCFAs to other molecules. For example, a considerable amount of acetate can be converted to butyrate through microbial cross-feeding [95]. The amount and range of SCFAs produced is completely dependent on the dietary fibre type [83], as well as the species of microbiota present in the colon [96]. Thus, the response of individuals to dietary fibre from a microbial and metabolic perspective is highly individualised and varied. The vast majority of SCFA produced are passively absorbed through the colonic lumen. It is estimated that only a minor portion (5–10%) of microbially produced SCFA are excreted in faeces [88]. Generally, the circulating concentration of SCFAs is relatively low, but it is well-established that these potent signalling molecules regulate numerous important processes throughout the body, particularly in the gut [21]. These metabolites are the main mechanism by which dietary fibre exerts its positive impact on host health and metabolism. Reductions in dietary fibre lead to lower microbiota diversity, particularly of species that are specialised in fermentation and thus SCFA production.

## 6. Physiological Effects of Dietary Fibre Mediated through SCFA Production

Microbially produced SCFAs, particularly butyrate, can activate a range of G-protein-coupled receptors (GPCRs) throughout the GIT, thereby influencing key metabolic processes such as glucose and lipid metabolism, as well as the regulation of satiety, all of which are implicated in the pathogenesis of obesity and metabolic syndrome [18]. The main GPCRs activated by SCFAs in humans are GPR41, GPR43 and GPR109A, which are expressed throughout the gut [97,98,99]. GPR41 and GPR43 are activated by SCFAs in the order of potency propionate ≥ butyrate > acetate [100,101], while GPR109A is only activated by butyrate [102,103]. In addition to receptor activation, SCFAs can regulate gene expression by epigenetic alterations to chromosomal DNA [104]. Butyrate and to a lesser extent propionate are known to inhibit histone deacetylase (HDAC) activity and expression, thus increasing histone acetylation [104]. Thus, there are two main mechanisms by which SCFA are thought to affect the biological response of a host [105]. 

Another important physiological effect of dietary fibre intake that indirectly affects the metabolic state is satiety. Both butyrate and propionate promote the expression of gluconeogenesis genes, which promotes satiety through the upregulation of hepatic portal vein glucose sensors [106,107]. Acetate can increase satiety through hypothalamic signalling [108]. Furthermore, through the activation of GPR41 and GPR43, SCFAs have the ability to induce the secretion of peptide YY (peptide tyrosine tyrosine) and GLP-1 (glucagon-like peptide 1), respectively, from enterendocrine cells situated in the colon, which are then released into systemic circulation [109,110]. Once in circulation, they can signal through the central nervous system and function to delay gastric emptying, thus delaying gut motility and prolonging the nutrient absorptive capacity [111,112,113,114]. Thus, it can be said that dietary fibre has an indirect effect on metabolism by reducing energy intake through its SCFA fermentation products, which act to regulate satiety via receptor-dependent and independent mechanisms.

## 7. Dietary Fibre in the Regulation of Glucose and Lipid Metabolism

Dietary fibre has been shown to positively regulate glucose metabolism in humans through SCFAs produced from fibre fermentation in the colon. Glucose metabolism is tightly regulated by the levels of the hormone insulin [115], and impairments in this regulation have been linked to a variety of diseases including obesity, T2DM, metabolic syndrome and CVD [116]. There are clear associations between the long-term consumption of fibre and reduced cardiovascular disease risk and T2DM [117]. SCFAs activate GPR43 in the colon and stimulate the production and secretion of GLP-1, which can directly promote insulin and inhibit glucagon secretion via interactions with pancreatic β-cells [112]. Additionally, it has been demonstrated that GLP-1 can improve β-cell receptivity to glucose, even in β-cells that are glucose-resistant [118,119]. Furthermore, propionate is mainly utilised for hepatic gluconeogenesis [105]. However, this SCFA has also been shown to stimulate β-cell insulin secretion in vitro, independently of the GLP-1 mechanism described [120]. It has also been demonstrated that SCFAs can inhibit hepatic gluconeogenesis by upregulating intestinal gluconeogenesis [107]. Reduced hepatic gluconeogenesis is clinically important as it is associated with insulin resistance and the development of some chronic gut diseases [121,122]. There are numerous studies that have tested the impact of SCFAs specifically on glucose metabolism through supplementation or administration independent of fibre [123,124]. Butyrate, independently of GPR41 and GPR43, can induce the expression of key intestinal gluconeogenesis genes phosphoenolpyruvate carboxykinase 1 and glucose-6 phosphatase [107], which is known to occur via an increase in cAMP in erythrocytes [125,126]. This indicates that SCFAs possess the power to positively alter glucose metabolism through de novo glucose synthesis. This in turn is detected by portal vein glucose sensors and initiates a signalling cascade, increasing insulin sensitivity and glucose tolerance. Furthermore, increases in circulating levels of peptide YY and GLP-1 were observed after rectal and intravenous administration of acetate in healthy humans [127]. SCFAs produced from fibre fermentation may influence lipid metabolism in the liver through GPR41 and GPR43 [128]. SCFAs can also activate PPAR-γ (peroxisome-proliferator activated receptor), which has a key role in the regulation of lipid metabolism [129]. Moreover, acetate can be used in the liver for lipogenesis [88]. Interestingly, the administration of acetate and propionate in humans led to a significant reduction in circulating fatty acid concentrations [130]. This is thought to be a GPR43-dependent mechanism causing the decrease in cellular lipolytic activity [131]. Furthermore, propionate can enhance the activity of adipose tissue lipoprotein lipase [132]. In humans, SCFA administration has also been shown to improve metabolic parameters in metabolically challenged individuals [133]. Canfora and colleagues (2017) [133] found that SCFA mixtures colonically infused in the concentrations and ratios reached after dietary fibre intake to overweight and obese men increased fat oxidation and energy expenditure, while lipolysis was decreased. Thus, it is clear that SCFA administration in the preclinical and clinical setting has highly beneficial effects on glucose and lipid metabolism. 

Experiments conducted in order to fully characterise the effect of RS on glycaemic, insulinaemic, lipidaemic and cholesterolaemic control in humans have obtained conflicting results [134,135]. RS2 consumption in insulin-resistant individuals had no effect on body weight or fat storage, although improvements in insulin sensitivity were reported [136]. Another study conducted in healthy individuals with RS2 found significant reductions in visceral and subcutaneous abdominal adiposity, which were accompanied by reductions in total cholesterol and LDL [48]. Work carried out by Haub et al. (2010) [137] also demonstrated this effect using both RS2 and RS4. Bodinham and colleagues (2010) [138] reported that RS2 had no significant effect on plasma glucose, but significantly lowered postprandial insulin levels in healthy adults. Other experiments highlighted how RS4 [139] and a genetically modified rice enriched with RS [140] substantially reduced glucose and insulin levels. Elsewhere, RS intervention has been demonstrated to improve insulin sensitivity [136,141]. Meanwhile, investigations into the effect of RS4 in healthy adults highlighted a 30% decrease in postprandial plasma glucose concentrations [142]. 

Arabinoxylan is the only type of hemicellulose that is extensively studied in relation to glucose metabolism and shown to lower postprandial glucose and insulin, thereby improving glycaemic response in healthy individuals [143] and patients with T2DM (Table 1) [144]. Kjølbæk et al. (2019) [145] recently described that while arabinoxylan increased butyrate production, the parameters of metabolic syndrome relating to glucose metabolism were unaltered. Another study conducted using this fibre subtype in a cohort of overweight individuals demonstrated no changes to glucose metabolism, although the microbiota were modulated and the abundance of butyrate-producing microbes was increased [50]. Lastly, arabinoxylan did not alter the lipid metabolome in overweight individuals [50,145]. 

Gum fibres have been demonstrated to cause a reduction in fasting blood glucose levels and insulin release in humans [146]. It is thought that these effects are related to the viscosity of gum fibre types, which allows for a profound bile acid-binding capacity [146]. Moreover, this property reduces glucose absorption through the intestinal lumen, thus reducing the access of amylase and other digestive enzymes to their substrates [147,148]. Gum fibres have been associated with increases in propionate [79] and butyrate [78,149] in the colon. No alterations to TAG or the expression of several key lipogenic genes were recorded, including fatty acid synthetase (FAS), acetyl CoA carboxylase and HMG-CoA reductase, as well as the key fatty acid receptor PPAR-γ [150]. A clinical trial involving T2DM patients who were supplemented with gum guar exhibited reduced levels of plasma FAs, HbA1c and waist circumference (Table 1) [151]. Interestingly, another study showed that gum guar increased the expression of hepatic sterol regulatory element binding protein 2 (SREBP2) and the LDL receptor [152,153]. The transcription factor SREBP2 tightly regulates cholesterologenesis and plays a key role in the activation of this metabolic process [154]. Recently, another investigation into the effects of gum guar in healthy individuals revealed significant reductions in postprandial glucose and insulin, in addition to lower TAG and LDL [155]. GOS, FOS and inulin have all been reported to have health-promoting effects through alterations to glucose and lipid metabolism [156]. A dietary fibre intervention to determine the effect of GOS and FOS supplementation reported that both led to a reduction in butyrate and total SCFA levels, an effect that was accompanied by detrimental effects on glucose metabolism [157]. The supplementation of GOS to obese, prediabetic males was reported to have no effect on insulin sensitivity, body composition or faecal SCFA levels [124], while FOS supplementation caused no significant alterations in glucose metabolism in healthy individuals [158]. However, it is likely that FOS positively regulates glucose metabolism, and that rather discrepancies in trial design are to blame for negative results [159,160]. It has been demonstrated that increases in inulin can reduce the levels of circulating triacylglycerides [161] and reduce plasma LDL [157]. Meanwhile, the administration of an inulin-propionate ester that is actively converted to free propionate in the GIT was shown to enhance peptide YY and GLP-1 secretion, with the long-term effect being reduced weight gain [162]. Another fibre intervention study with inulin in obese children reduced their overall fat mass (Table 1) [163]. Furthermore, it was found that the administration of inulin reduced body weight gain and improved glucose metabolism [164]. 

β-glucans have been reported as beneficial for the treatment, prevention and management of obesity, metabolic syndrome, T2DM and CVD [165]. It has been widely reported that β-glucan supplementation causes reductions in cholesterol and triacylglycerides in humans [166,167,168,169,170,171]. Othman and colleagues (2011) [172] highlighted that oat β-glucan may reduce LDL and total cholesterol in normal and hypercholesteraemic subjects. Oat-derived β-glucan, which has a profound ability to bind bile acids, can lower the postprandial insulin response, thus having a positive effect on glucose metabolism by reducing nutrient absorption [173,174]. Another study observed increased β-glucan consumption in healthy individuals, which improved glucose metabolism [175]. The main mechanism responsible for such effects has been attributed to the profound ability of β-glucan to sequester bile acids in the small intestine, increasing their excretion in faeces and increasing cholesterol-derived bile acid production in the liver [176,177]. A recent meta-analysis of the literature investigated β-glucan in diabetic individuals and determined that it can improve blood glucose levels and lipid parameters [178]. Another meta-analysis of this fibre subtype described similar effects in addition to significant reductions in TAG [179].

Several studies conducted using rye-based porridge and breads found that these products lowered hunger and the desire to eat when compared with wheat bread [180,181,182,183]. Furthermore, an intervention in a Swedish cohort led to reductions in postprandial glucose and insulin levels [184] (Table 1), while similar results were observed in a group of healthy individuals [183]. Another study reported that rye was able to reduce cholesterol levels in males with metabolic syndrome [185], while Leinonen et al. (2000), reported that rye fibre lowered LDL in males but not females [186].

Thus, dietary fibre can positively alter glucose metabolism in humans independently of colonic SCFA production [187]. By modulating intestinal transit time and delaying gastric emptying, reductions in rate of glucose absorption across the small intestinal lumen can be observed and are thought to contribute to the improvement of glycaemic control observed in clinical fibre intervention studies [188]. Almost all fibre subtypes mentioned in this section were able to reduce fasting and postprandial glucose and insulin levels. Similarly, all mentioned fibre types had a profound effect on reducing circulating cholesterol, LDL and TAG. Thus, it is clear that through the modulation of the gut microbiota and subsequent increases in SCFA production by fermentation, dietary fibre can positively impact glucose and lipid metabolism. This highlights the value of fibre in future nutraceutical interventions to combat ill metabolic health.

**Table 1 nutrients-13-01655-t001:** Different types of dietary fibre cause differential changes to the gut microbiota and metabolic status.

Author	Study Design	(n)	Study Population	Age (Years)	Duration (Weeks)	Fibre Type	Dose (g/day)	Microbiota	Metabolic Marker
Venktaraman, 2016 [45]	Fibre	20	Healthy	19–20	3	RS2	24	↑ *Ruminococcus bromii*, *Bifiodbacterium adolescentis*	↑ Butyrate
Zhang, 2019 [48]	Randomized, Double-Blind	19	Healthy	18–55	4	RS2	40	↑ *Ruminococcaceae_UCG-005* ↓ *Coprococcus*, *Bacteroides*	↓ Body Fat (%), LDL ↑ GLP-1, Acetate
Benítez-Páez, 2019 [50]	Randomized Crossover	30	Obese	36–52	4	Arabinoxylan	10	↑ *Prevotella*	↑ SCFA
Lu, 2004 [144]	Randomized Crossover	15	T2D	30–74	5	Arabinoxylan	15	Not measured	↓ Fasting glucose and insulin
Nicolucci, 2017 [163]	Double-Blind, Placebo	22	Obese	7–12	16	Inulin	10	↑ *Bifidobacterium*	↓ Body Weight, Body Fat (%), TAG
Ramirez-Faris, 2008 [57]	Randomized, Crossover	12	Healthy	30–64	3	Inulin	5	↑*Faecalibacterium prausnitzii*	Not measured
Wang, 2016 [66]	Randomized, Crossover	30	Metabolic Syndrome	27–78	5	ß-glucan	5	↑ *Bacteroides*, *Prevotella* ↓ *Dorea*	↓ Total Cholesterol
Mitsou, 2010 [69]	Randomized, Double-Blind, Placebo	52	Healthy	39–70	4	ß-glucan	1	↑ *Bfidobacterium*, ↓ *Bacteroides*	No significant change
Lappi, 2013 [85]	Randomized, Parallel	52	Metabolic Syndrome	40–65	12	Rye	75	↑ *Collinsella, Clostridium XIV* Bacteroides	No significant change
Lee, 2016 [184]	Randomized, Crossover		Healthy	18–60		Rye	40	Not measured	↓ Postprandial Glucose and Insulin
Vitaglione, 2015 [63]	Randomized, Parallel, Placebo	80	Obese	19–67	8	Wheat	70	↑ *Firmicutes*, *Bacteroidetes* ↓ *Clostridium*	No significant change
Dall’Alba, 2013 [151]	Randomized, Parallel	44	T2D	32–75	6	Gum guar	10	Not measured	↓ Waist Circumference, HbA1c, TAG

## 8. Obesity and Type II Diabetes Mellitus

Obesity has reached epidemic levels worldwide, is increasing due to the Western diet and poor lifestyle habits, and is a condition that is generally associated with excessive fat accumulation, insulin resistance and chronic low-grade inflammation [189,190]. This disease is highly associated with metabolic syndrome, which is characterised by insulin resistance, dyslipiemia, increased waist circumference, glucose intolerance and hypertension [191]. Both obesity and metabolic syndrome are closely linked and both are further associated with T2DM and CVD [192]. It is thought that the majority of parameters linked with metabolic syndrome can be attenuated by dietary fibre [193]. Interestingly, a study of healthy individuals found that weight gain was inversely correlated with the consumption of dietary fibre over time, thus demonstrating that fibre has a role in limiting weight gain in the long-term [194]. High dietary fibre consumption is associated with increased gut microbiota diversity and lower long-term weight gain [195]. A clear association exists between reduced microbial diversity and the prevalence of obesity, which likely reflects reduced fibre intake in these individuals [196]. The anti-obesity effects of dietary fibre have been widely described in the literature [150,197,198,199].

Reduced microbiota diversity and alterations to gut microbial structure and composition have been implicated in obesity [200,201]. It was shown that the relative proportion of *Bacteroidetes* phyla were significantly reduced in obese individuals compared with their lean counterparts. This abundance of *Bacteroidetes* was found to increase as obese individuals lost weight during the consumption of two different types of low-calorie diet [201]. Furthermore, faecal microbial transplantation from healthy mice to germ-free can increase the latter’s body weight as well as promoting insulin resistance without increases in caloric intake [202]. The same author also demonstrated how gut microbes promote the absorption of monosaccharides from the intestinal lumen, which subsequently increases hepatic lipogenesis, thus providing evidence that the human gut microbiome is a significant factor in contributing to increased nutrient utilisation accompanied by increased adiposity. This finding was further corroborated by Turnbaugh et al. (2006) [203] who demonstrated that the colonisation of healthy mice with “obese microbiota” significantly increased body fat composition as compared with mice who were colonised with “lean microbiota”. Major differences in the core microbiota between obese twins and lean twins were also demonstrated by this group, further highlighting that gut microbiota are altered in obesity and contribute to the disease phenotype [204]. The reduced *Bacteroidetes/Firmicutes* ratio reported here for obese people has also been described elsewhere [205,206,207]. Numerous other studies have failed to report the changes observed by Ley and colleagues on the *Bacteroidetes/Firmicutes* ratio in both human and animal studies [208,209,210,211,212,213]. At the species level, associations have been made between individual microbes and the development and progression of obesity and metabolic syndrome. For example, *Lactobacillus* has been linked to obesity, specifically by one group, as it was found that obese individuals had a significantly higher abundance of this bacterial genera than anorexic people [205]. Interestingly, a study conducted on obese children showed a higher abundance of *Faecalibatcerium prausnitzii* as compared with normal weight infants [210], contrary to findings in adults [205]. The colonisation of germ-free mice with *Bacteroides theaiotamicron* resulted in a 23% increase in body fat composition [202]. Nadal et al. (2009) [214] described reductions in *Clostrdium histolyticum*, *Eubacterium rectale* and *Clostridium coccoides*, but increases in the *Bacteroides*/*Prevotella* group, among obese individuals who lost more than 4 kg after a calories restricted intervention. In addition, Kalliomäki et al. (2008) [215] reported that higher levels of Staphylococcus aureus among infants were correlated with those individuals becoming obese. In the same study, *Bifidobacterium* species were observed to be in higher abundance in infants who remained normal weight. Furthermore, microbes that produce lipopolysaccharide (LPS) have been implicated in obesity. This was first characterised in mice, as it was found that obese animals had 2–3 times higher concentrations of circulating LPS [164,216]. It is clear that the gut microbiota are central to energy homeostasis in obesity [217]. As gut microbiota composition and metabolic phenotype differ between obese and lean individuals [218], dietary fibre intervention may represent an important strategy to tackle obesity. Indeed, it has been reported that differences between the obese and lean microbiome are similar to that reported between individuals consuming a high- versus low-fibre diet [26], further validating this hypothesis.

T2DM is strongly associated with both obesity and metabolic syndrome; in fact, it is estimated that over 80% of individuals with T2DM are overweight or obese, while increased weight gain is one of the main risk factors for the development of this disease [219]. Individuals with metabolic syndrome who suffer from varying degrees of insulin resistance are also highly likely to develop T2DM. Insulin resistance worsens over time unless measures are taken to prevent it, and eventually pancreatic β-cells can cease functioning and permanently stop producing insulin [220]. Qin et al. (2012) conducted one of the first major microbiota studies in individuals with T2DM. Interestingly, it was found that a cohort of Chinese patients with T2DM had a much lower abundance of well-established butyrate-producing microbes, such as *Roseburia intestinalis* and *Faecalbacterium prausnitzii*, as compared with healthy controls [220]. Moreover, the diseased group contained a much higher abundance of pathogenic bacteria, such as *Desulfovibrio*, *Clostridiales* and *Bacteroides caccae* [220]. Another study comparing Danish T2DM patients to healthy controls highlights significant differences in microbiota composition [221]. The authors reported significantly lower levels of the phylum *Firmicutes* and class *Costridia* in patients with T2DM, whilst also showing an alteration to the *Bacteroidetes*/*Firmicutes* ratio [221]. Further evidence of intestinal microbiota alterations in T2DM revealed statistically significant decreases in the populations of *Roseburia intestinalis* and *Faecalbacterium prausnitzii* in a cohort of Swedish children [222]. Indeed, more recently, Zhao et al. (2019) reported a higher abundance of *Proteobacteria* genera in the T2DM microbiome compared to controls [223]. It is interesting to note that work carried out by Forslund and colleagues (2015) showed that one of the main T2DM medications, metformin, is a confounding factor in microbiota studies in T2DM patients, as it has a significant effect on microbe populations in the gut [224]. These findings were confirmed elsewhere, with metformin prescription in T2DM patients being associated with higher levels of *Akkermansia mucinphila* and *Escherichia coli*. [225]. Although the exact mechanism of metformin action remains unclear, evidence suggests its effect may be mediated through the gut microbiota [226]. Thus, it is clear that individuals with obesity and T2DM experience alterations to their gut microbiota that likely contribute to the disease phenotype [200,227]. Meanwhile, dietary fibre has the ability to modulate the gut microbiota, altering the niche environment to promote the growth of health-promoting SCFA-producing microbes over those associated with the metabolism of other dietary components, while also contributing to improvements in glucose and lipid metabolism, further highlighting its importance as a potential therapy for metabolically challenged individuals.

## 9. Discussion

This literature review describes how different types of dietary fibre can differentially influence gut microbiota composition and metabolic status through microbial SCFAs production. Although much work remains to be done to further understand the fibre–microbiota–metabolic axis, evidence in this review suggests dietary fibre may be used as a nutraceutical intervention in the treatment of metabolic ill health.

The human diet has been subject to vast changes over the last few hundred years, which is a crucial factor in explaining the large difference in the prevalence of chronic metabolic disease between developed and developing nations. The gap formed by reduced fibre consumption has been filled by the consumption of energy-dense, high-glycaemic load foods, which are a staple of the Western diet. Our ancestors are thought to have consumed 100 g of fibre a day [228]; in comparison, people from non-industrialised nations generally ingest up to 50 g a day [229], while those from industrialised Western nations tend to consume only 12–18 g a day [230,231,232]. This profound difference in habitual fibre intake, in addition to the high protein and fat content of the Western diet, is associated with the difference in gut microbiota composition and richness observed between developed and developing nations, which has been reported in several studies [26,27,30,33,34]. Furthermore, when comparing the microbiomes of individuals from different industrialised countries (USA, Denmark, Spain, France, Italy and Japan), there is little difference in structure and composition [36], while vegan and vegetarian individuals from these societies have a gut microbiome more closely resembling that of non-industrialised populations who have a high habitual dietary fibre intake [29] and low associated metabolic disease prevalence. Current recommendations for dietary fibre intake lie between 30 and 35 g per day for males and 25 and 32 g per day for females [20]. Western societies are still not adhering to these guidelines, and low fibre consumption remains a major public health issue. In fact, even today’s recommended daily fibre intake is much lower than that consumed by our ancestors. There is substantial evidence that diet–microbiome interactions are essential to the ability of dietary fibre to improve the metabolic health of individuals suffering from obesity and metabolic syndrome, as well as other chronic gut diseases [164,233,234,235]. 

Although numerous epidemiological studies have demonstrated associations between dietary fibre intake and general health [117,236], some inconsistencies have been reported [237]. The majority of studies conducted have used whole foods or foods altered to have an increased quantity of fibre. Dietary fibre incorporated into whole foods may function differently to the same fibre isolated and consumed on its own [237]. As described in this review, many previously conducted dietary fibre interventions used varying quantities of fibre among a range of different cohorts, which resulted in heterogeneous changes to gut microbiota composition and metabolic parameters. Moreover, the human gut microbiome is highly individualized, with large variation evident in individuals within the same cohort [45], while metabolic responses are also diverse [238]. For example, a dietary fibre intervention is likely to have no effect in an individual whose microbiome does not include keystone species or other genera that encode the enzymatic machinery to degrade that specific fibre type [105].

All of these factors indicate that a more personalised approach may be required to tackle the global metabolic health issue. However, it is also plausible that the doses used in the majority of fibre interventions are too low to observe major effects on the gut microbiota or metabolic parameters, given that people from non-industrialised nations who are relatively free of metabolic disease consume around 50 g per day [229]. An important study conducted by O’ Keefe et al. (2015), wherein African Americans consumed a high-fibre (55 g per day), low-fat native South African diet for two weeks [229], showed major changes to gut microbiota composition as well as increased butyrogenesis and reduced risk of colorectal cancer [229], thus giving more weight to this hypothesis. Although it will be challenging to increase habitual dietary fibre intake to these heights, it is realistic. As suggested by Deehan and Walter (2016), fibre types such as RS, arabinoxylan and gum acacia are much better tolerated at higher doses, while gradual titration would allow the host to adapt sufficiently [232].

However, dietary fibre intervention alone may not be enough to reverse ill metabolic health. It was demonstrated using germ-free mice transplanted with a human faecal microbiome that certain taxa that increase in abundance upon increased fibre intake are not effectively transferred to the next generation [235]. It was also shown that a low-fibre diet dramatically altered the microbiome in just three generations, a change that could not be restored by increasing dietary fibre alone. The authors hypothesised that many of these fibre-fermenting taxa may have been permanently lost from the human gut microbiome, and in order to utilise dietary fibre as a treatment for metabolic disease, fibre intervention would have to be accompanied by faecal microbial transplant of bacterial taxa proficient in degrading fibre. 

In conclusion, evidence suggests that increased dietary fibre consumption can positively influence metabolic health by altering the gut microbiota. Dietary fibre may affect gut microbiota composition and function through the enrichment of only a few species that are able to adapt to an environmental change in the ecosystem, as they are enzymatically equipped to carry out fermentation. The increased abundance of fermenting species may increase the production and availability of SCFA, which in turn can reduce hyperlipidaemia, hyperglycaemia, hyperinsulinemia and hypercholesterolemia in a wide range of cohorts, both healthy and metabolically challenged. Although much work remains to be done to further understand dietary fibre–microbiome interactions, this remains the most promising and cost-effective method to reduce the burden of metabolic disease, and it is imperative that habitual fibre intakes are increased to current recommendations in Western societies. A greater understanding of how different types of dietary fibre influence gut microbiota composition and gut metabolome, as well as the biological mechanisms at play that influence host physiology, will be required to devise future recommendations regarding dietary fibre intervention as an adjunct therapy to treat metabolic disease.

## Figures and Tables

**Figure 1 nutrients-13-01655-f001:**
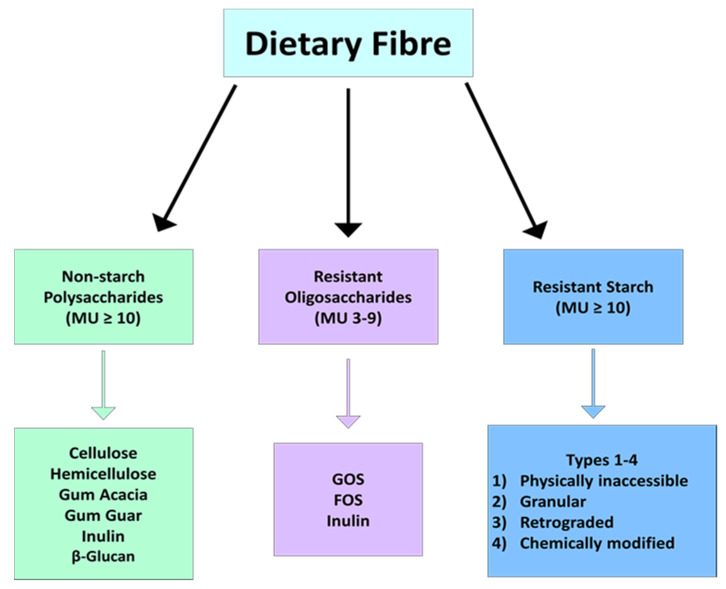
Schematic highlighting the different types of fibre divided by number of monomeric units. Abbreviations: monomeric units (MU), galacto-oligosaccharide (GOS) and fructo-oligosaccharide (FOS).

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
