# Peer review of "Dietary Fibre Modulates the Gut Microbiota"

_nutrients, 2021, doi:10.3390/nu13051655_

Round 1

Reviewer 1 Report

Dear Authors,

IAfter the review process, I have several comments: you should include in sections 7&8 more data about comparative fingerprinting of the human microbiota in diabetes and cardiovascular disease; in addition, you should include comments about the importance of fiber to combat oxidative stress through microbiota modulation - this aspect represents a new research direction in the valorization of microbiota modulation in the case of degenerative pathologies.

Best regards!

Author Response

We wish to thanks reviewer #1 for their comments. More information has been included in section 8 regarding the human microbiota, as requested. We have purposely kept the focus of this review on the effects of dietary fibre on metabolism. We are aware that fibre is also associated with anti-inflammatory and protective effects against ROS however this is outside the scope of the current review.

Reviewer 2 Report

This narrative review examines how different types of dietary fibre can differentially alter glucose and lipid metabolism through changes in gut microbiota composition and function. 

The authors devoted much of their work to presenting the different types of fibre and their effect on the microbiota. Only two paragraphs focused on the effects on type 2 diabetes and lipid status. 

The conclusions of the review are rather general and do not offer particular insights into clinical practice or new research in this area. In fact, the authors write that: "There is clear evidence indicating that increased dietary fibre consumption can positively influence metabolic health by altering the gut microbiota. The increased abundance of fermenting species will increase the production and availability of SCFA which in turn will reduce hyperlipidaemia, hyperglycaemia, hyperinsulinemia and hypercholesterolemia in a wide range of cohorts both healthy and metabolically challenged."

I think that in order to aspire to publish the work in a journal of the level of Nutrients, the authors should upgrade the review.

It would be useful to include:

  1. the criteria used to select the articles (keywords, database, etc.)
  2. The PRISMA flow chart
  3. A summary table with the results of the most significant papers
  4. a more extensive discussion and conclusions.

Author Response

We wish to thanks reviewer 2 for their comments.

Reviewer 2 wrote “The authors devoted much of their work to presenting the different types of fibre and their effect on the microbiota. Only two paragraphs focused on the effects on type 2 diabetes and lipid status.” –

Information about the effect of fibre on T2DM and lipid status is present throughout the review. Section 4 focuses on how different types of dietary fibre have been shown to alter gut microbiota composition in many different cohorts of metabolic disease. Similarly, Section 7 gives a detailed overview of how these different types of fibre influence both glucose and lipid metabolism and again includes studies conducted on different cohorts of metabolic disease including T2DM. The focus of Section 8 is on how the microbiome is altered by obesity and T2DM disease states rather than examining specific fibre interventions (already addressed in other sections of the review). We specifically want this review to encompass how the different types of dietary fibre can differentially alter the gut microbiome and metabolic health in general. As a result, we did not break it down in a disease specific manner.

Reviewer 2 also wrote “The conclusions of the review are rather general and do not offer particular insights into clinical practice or new research in this area”.

Discussion and conclusion have been amended accordingly.

Regarding reviewer 2 comments about including PRISMA flow chart, criteria used to select articles and a summary table. This manuscript was never intended to be a systematic literature review therefore use of a methods outlined by reviewer 2 (PRISMA flow chart, criteria used for inclusion, summary tables) were not employed (as they are not appropriate in this context).

Round 2

Reviewer 1 Report

Dear Authors,

I do not have other comments.

Best regards.

Author Response

We would like to thank the reviewer for their contribution.

Reviewer 2 Report

There were no major changes compared to the first version of the paper. I think the changes are necessary and I reiterate them below.

This narrative review examines how different types of dietary fibre can differentially alter glucose and lipid metabolism through changes in gut microbiota composition and function. 

The authors devoted much of their work to presenting the different types of fibre and their effect on the microbiota. Only two paragraphs focused on the effects on type 2 diabetes and lipid status. 

The conclusions of the review are rather general and do not offer particular insights into clinical practice or new research in this area. In fact, the authors write that: "There is clear evidence indicating that increased dietary fibre consumption can positively influence metabolic health by altering the gut microbiota. The increased abundance of fermenting species will increase the production and availability of SCFA which in turn will reduce hyperlipidaemia, hyperglycaemia, hyperinsulinemia and hypercholesterolemia in a wide range of cohorts both healthy and metabolically challenged."

I think that in order to aspire to publish the work in a journal of the level of Nutrients, the authors should upgrade the review.

It would be useful to include:

  1. the criteria used to select the articles (keywords, database, etc.)
  2. The PRISMA flow chart
  3. A summary table with the results of the most significant papers
  4. a more extensive discussion and conclusions.

Author Response

(The authors gave the same response as above.)
